# Nitric-Oxide-Mediated Signaling in Podocyte Pathophysiology

**DOI:** 10.3390/biom12060745

**Published:** 2022-05-25

**Authors:** Marharyta Semenikhina, Mariia Stefanenko, Denisha R. Spires, Daria V. Ilatovskaya, Oleg Palygin

**Affiliations:** 1Division of Nephrology, Department of Medicine, Medical University of South Carolina, Charleston, SC 29425, USA; semenikh@musc.edu (M.S.); stefanen@musc.edu (M.S.); 2Department of Physiology, Medical College of Georgia, Augusta University, Augusta, GA 30912, USA; despires@augusta.edu (D.R.S.); dilatovskaya@augusta.edu (D.V.I.); 3Department of Regenerative Medicine and Cell Biology, Medical University of South Carolina, Charleston, SC 29425, USA

**Keywords:** nitric oxide synthase, glomerulus, lupus nephritis, hypertension

## Abstract

Nitric oxide (NO) is a potent signaling molecule involved in many physiological and pathophysiological processes in the kidney. NO plays a complex role in glomerular ultrafiltration, vasodilation, and inflammation. Changes in NO bioavailability in pathophysiological conditions such as hypertension or diabetes may lead to podocyte damage, proteinuria, and rapid development of chronic kidney disease (CKD). Despite the extensive data highlighting essential functions of NO in health and pathology, related signaling in glomerular cells, particularly podocytes, is understudied. Several reports indicate that NO bioavailability in glomerular cells is decreased during the development of renal pathology, while restoring NO level can be beneficial for glomerular function. At the same time, the compromised activity of nitric oxide synthase (NOS) may provoke the formation of peroxynitrite and has been linked to autoimmune diseases such as systemic lupus erythematosus. It is known that the changes in the distribution of NO sources due to shifts in NOS subunits expression or modifications of NADPH oxidases activity may be linked to or promote the development of pathology. However, there is a lack of information about the detailed mechanisms describing the production and release of NO in the glomerular cells. The interaction of NO and other reactive oxygen species in podocytes and how NO-calcium crosstalk regulates glomerular cells’ function is still largely unknown. Here, we discuss recent reports describing signaling, synthesis, and known pathophysiological mechanisms mediated by the changes in NO homeostasis in the podocyte. The understanding and further investigation of these essential mechanisms in glomerular cells will facilitate the design of novel strategies to prevent or manage health conditions that cause glomerular and kidney damage.

## 1. NO Signaling in the Glomerular Podocyte

Podocytes are specialized epithelial cells that wrap around the glomerular capillaries and are integral to glomerular structure and function [1,2]. Adjacent podocytes form a network of interdigitating foot processes representing a major center of this cell function–control of the renal filtration barrier. The barrier actively prevents the passage of large plasma proteins and selectively allows the movement of small molecules and water [2,3]. The podocytes’ foot processes form a special category of intercellular junctions named slit diaphragm, which uses actin cytoskeleton proteins to control filtration and initiate signaling pathways that regulate the plasticity of foot processes [4]. Podocytes are limited in their ability to proliferate; thus, during cardiorenal pathology and glomerular injury, these cells display hypertrophy, foot processes effacement, and detachment. Eventually, irreversible loss of podocytes from the glomerular capillary leads to deterioration of the glomerular filtration barrier, proteinuria, and glomerular disease [1,3,5]. Podocyte injury can be caused by multiple factors and is commonly associated with diseases that alter the microenvironment of the podocyte, such as hypertension and diabetes [2,5]. The nitric oxide (NO) is essential for glomerular filtration barrier homeostasis. It may interact with other vasoactive hormones, such as angiotensin II and endothelin-1, which play critical roles in developing renal pathologies [3,6]. NO is an intracellular and extracellular ubiquitous messenger molecule involved in various signaling pathways for a diverse set of physiological processes, including inflammation, immunity, ion channel modulation, gene transcription, and vascular tone [7,8]. NO is synthesized from L-arginine by a group of enzymes called NO synthases (NOS) [7,9]. There are three isoforms of NOS, often described based on their tissue expression: neuronal NOS (nNOS or NOS1), inducible NOS (iNOS or NOS2), and epithelial NOS (eNOS or NOS3) [8,10]. Despite the nomenclature, the expression of these enzymes is not limited to one tissue type. For instance, the kidney expresses all three isoforms that are known to affect natriuresis and diuresis and contribute to blood pressure control [9,11].

Nitric oxide (NO) is a vital molecule involved in a plethora of signaling functions in the kidney. NO can be released by a variety of renal cells, including endothelial and mesangial cells, macula densa, and podocytes [12,13]. Thus, NO release has the potential to directly affect the glomerular function and regulate glomerular ultrafiltration [9]. In addition, NO is essential for renal hemodynamics and blood pressure regulation and plays an important role in the modulation of diuresis tubuloglomerular feedback and sodium reabsorption [14]. It is well-established that NO is crucial for normal glomerular function and plays a protective role in preventing glomerular diseases. Podocytes are highly specialized cells of the glomerulus controlling permeability of the glomerular filtration barrier. Due to the structural composition of the slit diaphragm, podocytes cooperate with endothelial and mesangial cells to coordinate signals modulating cell adhesion and matrix interaction to maintain and control glomerular permeability. The precise regulation and organization of podocytes’ cytoskeletal architecture are essential to support the kidney filtration barrier. The direct relation of cytoskeletal organization with changes in NO production was previously shown for endothelial cells and may play an important physiological and pathophysiological role in podocyte function [15]. It was shown that podocytes can release NO in response to various receptor-mediated stimuli, including angiotensin II (Ang II) [6], which may implicate paracrine and autocrine regulation of podocytes and surrounding glomerular cells. Ang II is a crucial component of the renin–angiotensin system (RAS)-regulated vasoconstriction and blood pressure control in the body. It is well-known that Ang II could stimulate NO production in the kidney and promote vasodilation [16]. The activation of the Ang II pathway strongly impacts podocyte cytoskeletal remodeling, including focal adhesion, cell motility, and nephrin signal transduction [17,18]. We have shown that the application of Ang II on podocytes of the freshly isolated glomerulus from rats mediates the production of NO (Figure 1a). Ang II-mediated production of NO in podocytes is facilitated through the Ang II receptors cascade and triggers a rapid elevation in intracellular calcium [19,20]. The interaction mechanisms between NO and intracellular Ca^2+^ in podocytes require extensive investigation, and we predict that, similarly to vasoconstriction/vasodilation in endothelial cells, the observed Ang II-mediated NO release may regulate podocytes’ cytoskeleton dynamics. In aortal smooth muscles, the initial rapid decrease in intracellular calcium induced by NO is accounted for by the uptake of Ca^2+^ into intracellular stores by SERCA [21]; however, there are no data regarding the existence of this mechanism in glomerular cells. Ca^2+^/NO crosstalk is extensively studied in vascular physiology, but it is still unclear which NO synthases and downstream mechanisms are essential in regulating podocyte function.

The detection of NO production is typically monitored using fluorescent dye 4-Amino-5-Methylamino-2’,7’-Difluorofluorescein diacetate (DAF-FM) [22,23]. DAF-FM is a reagent that is used to detect and quantify a wide range of concentrations of nitric oxide (NO). It is essentially nonfluorescent and requires a chemical interaction with NO to form a fluorescent compound (dichlorofluorescein) [22]. Fluorescent methods of NO detection have high specificity; however, it was shown that the presence of O_2_^−^ or H_2_O_2_ may affect the fluorescence of DAF-FM, resulting in under- or overestimated NO production [24]. The specificity of H_2_O_2_-mediated NO release in different cells was proven by multiple methods, including a cell-trappable copper (II) fluorescent probe [25].

NO can also be released in response to extracellular hydrogen peroxide (H_2_O_2_) application in podocytes of freshly isolated glomeruli (Figure 1b). It was shown that H_2_O_2_ can stimulate NOS activation, and this signaling mechanism may be involved in the modulation of NO signaling responses by oxidative stress [25]. The release of NO in response to H_2_O_2_ stimulation was shown in many cell types, including cardiomyocytes and endothelial cells [26,27,28,29]. Interestingly, low doses of H_2_O_2_ may activate NOS and promote cellular NO synthesis. In contrast, high H_2_O_2_ treatment and oxidative stress conditions may lead to a decrease in NO bioavailability [6,26]. In addition to signaling molecule properties, NO can act as an antioxidant by scavenging other ROS and counteracting oxidative stress [30]. Under physiological conditions, NO diminishes mitochondrial superoxide production (O_2_^−^), the primary oxygen free radical produced in mitochondria [31]. Loss in NO bioavailability is associated with increased levels of O_2_^−^ [32]. The presence of NO donors helps scavenge O_2_^−^, but in some pathological conditions, the joint biochemical reaction between NO and O_2_^−^ may increase nitrosative stress [32]. This reaction produces peroxynitrite, which is normally involved in protein aggregation, turnover, signaling, and immunological processes in the cells. Under some pathological conditions, overproduction of NO may lead to nitrosative stress (Figure 2), and following peroxynitrite formation triggers several cytotoxic processes, including apoptosis, necrosis, and calcium dysregulation [33]. Increased nitrosative stress is associated with different renal pathologies, which suggests that correct reactive oxygen species (ROS) and reactive nitrogen species balance is essential for cellular health [33,34]. The imbalance of ROS–NO crosstalk can attenuate different pathological conditions such as hypertension, diabetes, and lupus nephritis [35,36,37]; thus, further investigation of this pathway is required for understanding pathophysiological processes in podocytes.

As a signaling molecule, NO can directly act through its receptor, soluble guanylate cyclase (sGC). sGC is a central component of the NO-signaling pathway, which is currently considered to be a promising pharmaceutical target in different pathologies [38]. sGC activators or stimulators have considerable therapeutic advantages compared to NO-donors, as using them allows avoiding the development of nitrosative stress [39]. In glomeruli, sGC is shown to be highly abundant in podocytes and some of the parietal cells but not in mesangial cells [40]. One of the most promising targets of sGC activity is an atrial natriuretic peptide (ANP), which can affect several renal functions, including sodium secretion, RAS, and glomerular filtration rate [41,42]. Interestingly, the knockout of *Nppa*, the ANP protein-coding gene, in the Dahl salt-sensitive rat background exacerbates glomerular injury and hypertension [43]. Natriuretic peptides/sGC signaling protects podocyte integrity under pathologic conditions, most likely by suppressing transient receptor potential canonical (TRPC) channels [44]. Indeed, both NO and sGC activation can decrease transient receptor potential canonical channel 6 (TRPC6) expression, inhibit TRPC6-mediated Ca^2+^ influx, and reduce podocyte injury [45]. TRPC6 is one of the key proteins responsible for calcium flux in the podocytes, and activation of the TRPC6 pathway is linked to multiple podocyte and glomerular diseases [46]. Furthermore, several reports indicate that NO stimulates cGMP synthesis in podocytes and prevents podocyte injury by inhibiting pathological TRPC6-mediated signaling [45,47]. On the contrary, for the TRPC5, also known as Ca^2+^ influx modulator in podocytes, NO mediates their activation and consecutively induces the entry of Ca^2+^ into cells by cysteine S-nitrosylation mechanism [48,49].

Despite the recent advances in the understanding of pathways triggering NO production in the podocyte, the physiological and pathophysiological ramifications of such effects are still unclear. NO mediates diverse signaling pathways, and more studies are needed to shed light on the mechanisms behind this effect.

## 2. Nitric Oxide Synthase Isoforms in the Glomerulus

Nitric oxide synthases (NOS) are a group of isoenzymes that promote NO production utilizing L-arginine as the substrate and molecular oxygen and reduced nicotinamide-adenine-dinucleotide phosphate (NADPH) as co-substrates [8]. There are three NOS isoforms: neuronal nitric oxide synthase (NOS1), inducible nitric oxide synthase (NOS2), and endothelial nitric oxide synthase (NOS3), encoded by *NOS1*, *NOS2*, and *NOS3*, respectively. Ca^2+^/calmodulin-dependent protein kinase is essential for the NOS phosphorylation and further NO-mediated signal transduction mechanism [52,53]. However, NOS subunits can also be activated by stimuli that do not require sustained increases in intracellular Ca^2+^. Fluid shear stress or hormonal factors such as insulin can promote NOS activation through protein kinase A (PKA) or serine/threonine-protein kinase Akt and the AMP-activated protein kinase (AMPK) [8].

The expression of NOS isoforms in kidney tissue under normal physiological conditions is similar for male and female subjects (Figure 3). However, several studies suggested that changes in NOS expression and corresponding mechanisms of NO signaling in males and females could be significantly different in pathological conditions [54]. Female spontaneous hypertensive rats (SHR) exhibit increases in the NO-sGC pathway in renal tissue, as well as significant elevation in NOS1 and NOS3 synthases expression in the inner medulla [55,56]. In contrast, Zucker diabetic rats’ kidneys show higher levels of NOS1 and NOS3 in males than in females [57]. Thus, possible sexual dimorphisms in the expression of NOS synthases in podocytes should be considered for future studies on the role of NO in glomerular pathology.

Studies have shown that all three NOS isoforms are present in the glomerulus in both human and rodent species (Susztaklab Kidney Biobank. Available online: https://susztaklab.com, Accessed on 15 May 2022). NO plays a significant role in endothelium-dependent relaxation of vessels, and NOS3 can be found in endothelial cells of glomerular capillaries [58]. However, only a few studies report individual expression of the NOS isoforms in specific glomerular cell types. In contractile mesangial cells, NO may cause relaxation and act as a proinflammatory mediator, which rapidly increases NO formation in response to inflammatory cytokines. It was shown that interleukin-1β (IL-1β) induces NOS2 expression and activation in mesangial cells [59]. The studies in mesangial cells indicate that antioxidant and pro-oxidant properties of NO may depend on the NOS2 activity and the availability of ROS [60]. The reports of NOS expression and their physiological function in podocytes are minimal and contradictory. The expression of NOS1 and sGC in human podocytes was confirmed by several experimental approaches [40,61]. Both RNAseq and immunohistochemistry indicate that under normal physiological conditions, podocytes presumably express NOS1 [40,62]. However, PCR-based studies also suggest the expression of NOS2 and NOS3 isoforms in podocytes [45]. Similar to mesangial cells, NOS2 expression in podocytes may rapidly change in response to immune stimuli and can play a crucial role in the development of inflammation-mediated podocytopathies [63]. In addition, NOS activity is involved in several physiological and pathophysiological processes of the podocyte. It was shown that NO production in podocytes is reduced during hypertension in male rats, which may promote further disease development [6]. The increase in NOS2 expression might play a negative role via direct involvement in podocyte pathophysiology. Systemic inflammation associated with lupus nephritis induces an increase in *Nos2* expression and reduces the levels of intrinsic inhibitors of *Nos2* transcription in glomerular cells [64,65]. NOS2-mediated peroxynitrite overproduction may further lead to podocyte damage by shifting the redox balance, as mentioned above. Overall, there is limited knowledge about isoform prevalence and possible remodeling of NOS expression during pathological conditions in podocytes and glomerular parietal epithelial cells, revealing a need for further investigation.

## 3. H_2_O_2_ and NO Crosstalk

Physiological concentrations of H_2_O_2_ are essential for maintaining the cellular redox homeostasis, and as a second messenger, H_2_O_2_ takes part in cell proliferation, survival, apoptosis, and phagocytosis [66]. It was shown that H_2_O_2_ may stimulate NOS3 and NOS2 isoforms of NO synthase and components of NADPH oxidase [67]. Thus, H_2_O_2_ can provoke NO release in the glomerulus and endothelial cells [6,27]. It was shown that H_2_O_2_-induced promotion of NOS activity and subsequent NO production mediated by cooperative effects between PI3-K/Akt-dependent NOS phosphorylation and activation of the MEK/ERK1/2 pathway [27,68] (Figure 4). In addition, in many cells, this signaling could be activated by Ang II-stimulated NADPH oxidase activity [69]. However, high H_2_O_2_ concentrations may result in a decrease in NO level, NOS3 activation, and Akt phosphorylation [27]. Moreover, high H_2_O_2_ levels may lead to oxidative stress and induce glomerular cell damage. H_2_O_2_-induced oxidative stress contributes to the development of a number of chronic kidney diseases, including hypertension, and may lead to a substantial decrease in NO bioavailability in glomerular podocytes [6].

NADPH oxidases mediate several cellular signaling processes and are involved in cell growth, differentiation, apoptosis, and fibrosis [74]. NADPH oxidases are considered to be the most powerful source of ROS production, which is their sole function [75]. NOX1, NOX2, and NOX4 are NADPH oxidase isoforms widely expressed in the glomerular cells [76]. In addition, podocytes express the Ca^2+^-regulated NOX5 isoform, which could, in some pathological conditions, exacerbate podocyte dysfunction, contributing to albuminuria and hypertension [77,78]. It was also reported that endothelial Nox5 promotes NO-cGMP signaling dysfunction and hypertension in mice [79]. Unlike other NOX isoforms, NOX4 is constitutively active, producing primarily H_2_O_2_, while other NOXs mainly produce O_2_^−^ [80]. Podocyte-specific Nox4 deletion attenuates albuminuria and diabetes-induced loss of nephrin expression [81]. In addition, NOX activity may be upregulated by prolonged Ang II exposure and high salt [82] and is linked to hypertension development. Overall, dysfunctions of NOX-mediated ROS production lead to oxidative stress and an associated decrease in NO [83]. Thus, further understanding of the role of NOX5 and NOX4 in the regulation of NO bioavailability in podocyte during the development of glomerular disease is required. 

## 4. Physiological and Pathophysiological Facets of the NO Pathway in Glomerular Podocytes

A variety of factors can affect NO bioavailability in the kidneys. NO modulates glomerular ultrafiltration and renal hemodynamics, and these mechanisms play a critical role in regulating tubuloglomerular feedback [84,85]. It is established that the synthesis of NO and the expression of the NOS isoforms in the glomerulus can substantially change during the development of renal injury [86]. NO’s role in regulating the glomerular filtration barrier was examined by measuring albumin permeability in isolated glomeruli. Interestingly, two independent reports indicate that the exposure of isolated glomeruli of male salt-resistant Sprague-Dawley and salt-sensitive Dahl SS rats to NO donors has a direct effect on the permeability barrier of glomerular tufts [6,87]. In both cases, preincubation with NO donors was proposed to affect NO level in podocytes and resulted in the absence of response to oncotic gradient change, possibly indicating impairment of the glomerular permeability barrier. However, participation of endothelial and mesangial cells in this process was not tested and thus cannot be excluded. 

Other findings indicate that NO is essential for glomerular filtration barrier maintenance. NOS inhibition by L-NAME caused an acute increase in glomerular permeability in male Wistar rats, which was reversed by the ROS antagonism or activation of the guanylyl cyclase-cGMP pathway [88]. It was suggested that NO exhibits a protective effect on glomerular permeability, conceivably by antagonizing ROS and acting through the NO/sGC/cGMP pathway, while inhibition of NO synthesis could enhance O_2_^−^-mediated oxidative injury under pathologic conditions [88,89]. Phosphodiesterase 5 (PDE5) is highly expressed in the glomerulus, and PDE5 inhibitors lead to increased or prolonged NO signaling due to potentiation of endogenous cGMP [13]. Clinical trials indicate that type 2 diabetic patients treated with PDE5 inhibitors significantly decreased urinary albumin excretion, one of the primary pathological markers of glomerulus damage [90]. In animal models of diabetes (lepr(db/db) male and female mice), the deletion of *Nos3* increases the urinary albumin/creatinine ratio and promotes diabetic nephropathy similar to human disease. [91]. Furthermore, NOS inhibition provoked albuminuria in diabetic and hypercholesterolemic patients [92]. Moreover, inhibition of NOS using L-NMMA resulted in increased urine albumin-to-creatinine ratio in hypertensive patients with type 2 diabetes [92]. It should be noted that L-NMMA shows low inhibitory potency toward NOS3 activity, suggesting that this effect may be connected to the other NOSs [93].

The abovementioned correlation of albuminuria and deficit in NO levels was further confirmed in experiments on immortalized podocyte culture. The combination of high glucose and low NO shifts podocyte actin dynamics towards actin depolymerization, which results in reduced podocyte motility [94]. Immortalized mouse podocytes exposed to high glucose also revealed decreased expression of myosin 9A (MYO9A), a calcium-regulated protein that acts as a processive motor along actin filaments, regulating epithelial cell–cell adhesion, cell morphology, migration, and membrane trafficking [94,95]. MYO9A also inhibits transforming Ras homolog family member A (RhoA) protein activity, while induction of RhoA in podocytes can cause albuminuria and foot processes’ effacement [96]. NO is involved in RhoA activity antagonism by multiple forms of crosstalk between the RhoA/ROCK pathway and the NOS/NO/cGMP pathway, suggesting that decreased NO levels and disrupted NO-RhoA interference during diabetes can cause podocyte damage and affect the integrity of glomerular filtration barrier [97]. Interestingly, HDAC1-mediated deacetylation was proposed to regulate NO production in the renal cells, and inhibition of Histone deacetylase 1 (HDAC1) activity in podocytes may suppress the progression of human proteinuric kidney diseases [98,99,100].

The reduced NO bioavailability is one of the important risk factors for hypertensive conditions. For example, the increase in blood pressure induced by high salt intake could be associated with the inability of salt-sensitive individuals to increase or sustain NO production [101]. Indeed, as we recently showed in an animal model of salt-sensitive hypertension, NO production in podocytes is significantly blunted and associated with albuminuria during the development of salt-induced blood pressure and renal damage [6]. The mechanisms of podocyte injury in hypertension could be related to the crosstalk between NO and calcium signaling and modulation of intracellular redox balance by the RAS.

The decreased NO synthesis plays a vital role in the relationship between albuminuria and hypercholesterolemia. It was suggested that hypercholesterolemia-mediated reduction in NO bioavailability directly promotes podocyte injury and is associated with a pathological remodeling in the structural and functional integrity of the glomerular filtration barrier [102].

It was shown that the increase in systemic NO production correlates with disease severity in human systemic lupus erythematosus (SLE) and lupus nephritis patients [103]. Several murine and human studies have associated lupus nephritis with podocyte specific increase in NOS2 expression [63,104]. The stimulation of podocytes with Toll-like receptor 4 (TLR4) agonist lipopolysaccharide (LPS) rapidly promoted an increase in NOS2-mediated superoxide production, permeability to albumin, and podocyte cell motility [63]. Interestingly, the knockout of *Nos2*^−/−^ did not prevent the pathology in the mouse model of proliferative glomerulonephritis [105]. However, proteinuria was further reduced in these mice by pharmacological inhibition of NOS, indicating possible involvement of other NOS subunits in this glomerular pathology.

## 5. Conclusions and Future Directions

Despite the established importance of NO signaling in podocyte physiology and glomerular function, the role of NOS-mediated mechanisms in these cells is still not fully understood. Recent reviews that focused on NO signaling in the kidney largely overlooked the presence of NO homeostasis in the glomerulus due to the lack of data and multiple controversial issues. Future studies should be aimed at revealing the glomerular expression patterns of NOS subunits in normal and pathological conditions, the crosstalk between the Ca^2+^ signaling and NO, and potential therapeutic targets for the treatment of podocyte dysfunction and associated glomerular diseases.

## Figures and Tables

**Figure 1 biomolecules-12-00745-f001:**
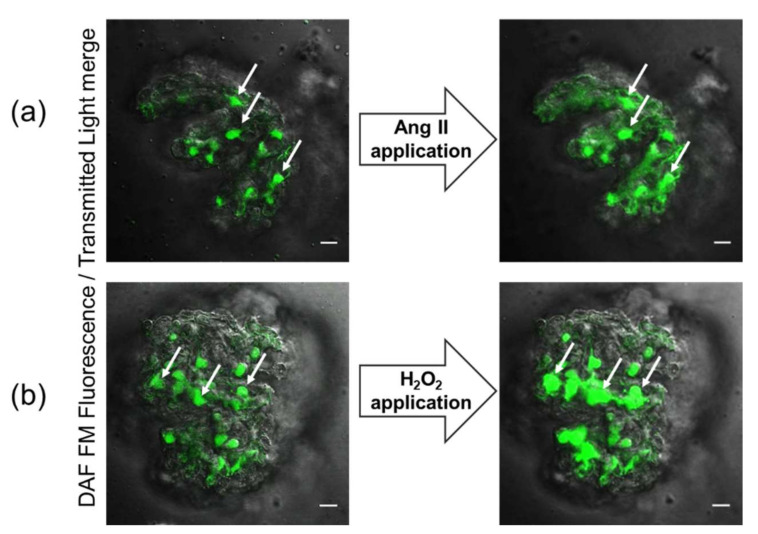
Merged fluorescent and transmitted light images of freshly isolated decapsulated glomeruli loaded with DAF-FM dye (ex. 488, em. 510/20 nm). Microphotographs show the surface localization of podocytes in the glomerulus. Confocal imaging of glomerulus before (left panels) and after (right panels) application of Ang II (**a**) or H_2_O_2_ (**b**) provoke elevation of intracellular NO levels in podocytes. Arrows denote podocytes discernable by their morphology and location. The scale bar is 20 μm.

**Figure 2 biomolecules-12-00745-f002:**
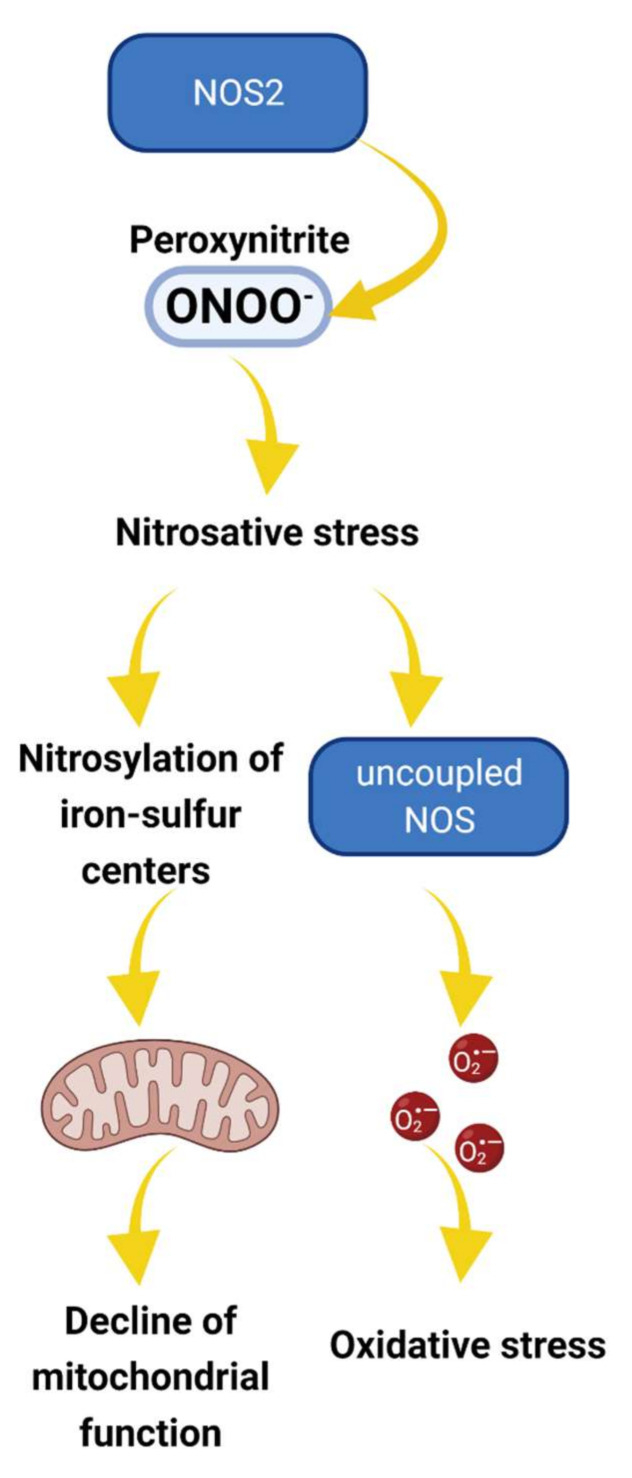
The proposed role of NOS2 in nitrosative stress and pathogenesis of podocyte and glomerular diseases. Under pathological conditions such as hyperglycemia, cellular production of NO and reactive oxygen species (ROS) are increased, promoting peroxynitrite (ONOO^−^) production [50]. Peroxynitrite and other reactive nitrogen species oxidatively inactivate different mitochondrial proteins, affecting the iron–sulfur centers and altering mitochondria function. Alternatively, nitrosative stress results in NOS uncoupling, further increasing O^2−^ production and oxidative stress [51]. Overall, the cascade of events may alter mitochondria activity and promote redox imbalance leading to irreversible damage and podocyte dysfunction.

**Figure 3 biomolecules-12-00745-f003:**
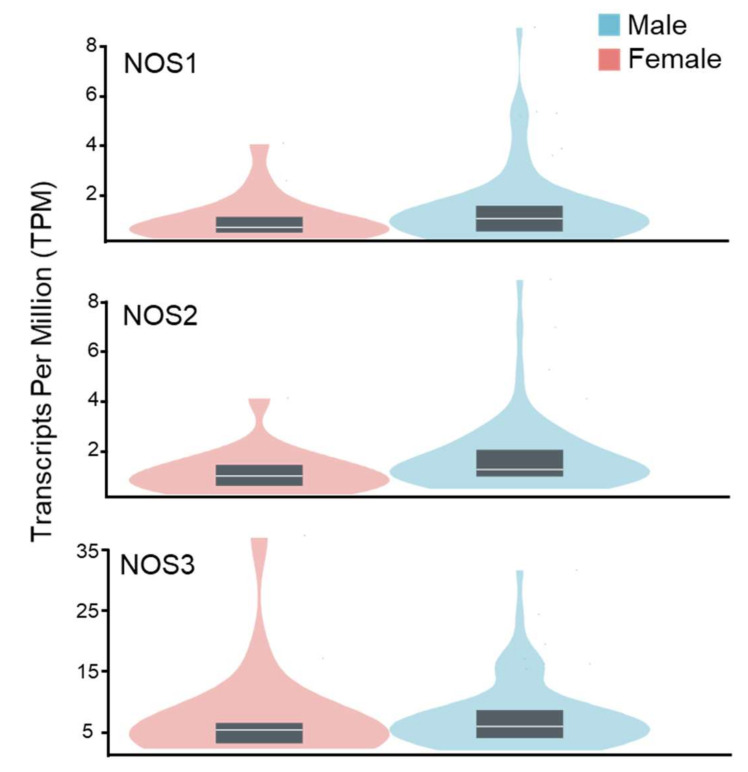
Genotype-tissue expression data for NO synthases in the human kidney cortex of males and females. Data were obtained from the GTEX portal (dbGaP Accession phs000424.v8.p2, 12 May 2022). Sample size: female n = 19; male n = 66.

**Figure 4 biomolecules-12-00745-f004:**
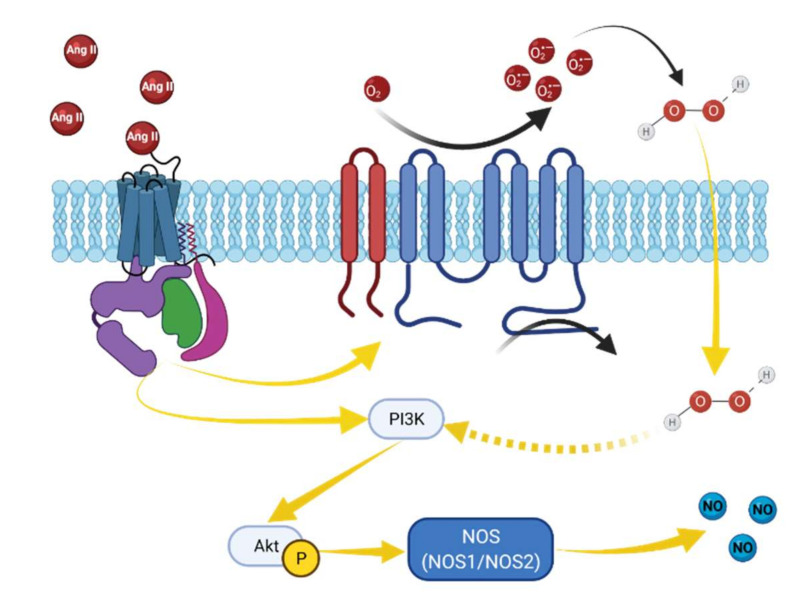
Proposed angiotensin II and H_2_O_2_-induced NO elevation in podocytes. Angiotensin-II metabolites may activate production of NO in cells through AT2R or Mas receptors [70,71]. Activation of this GPCR is followed by PI3-K/Akt-dependent NOS phosphorylation and subsequent NO production [71,72]. Angiotensin II binds to its receptors and activates NADPH oxidase, which in turn increases ROS generation. NADPH oxidases, such as NOX4, mediate H_2_O_2_ release, which may further promote PI3-K/Akt-dependent NOS phosphorylation and subsequent NO production [73].

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
