# Peer review of "Nitric-Oxide-Mediated Signaling in Podocyte Pathophysiology"

_biomolecules, 2022, doi:10.3390/biom12060745_

Round 1

Reviewer 1 Report

This is a well-written review descripting our current knowledge about NO signaling in podocytes under physiological and pathophysiological conditions. The review article highlights gaps in our understanding of NO function in podocytes, thus identifying areas for future research opportunities.

It is important to start the body of the review article by providing some background about NO as a signaling molecule and cellular functions and characteristics of podocytes, before discussing “NO signaling in the glomerular podocyte”.

Given the established variation reported in the abundance of NOS isoforms in kidneys from rats and mice, it is crucial to include the species in NOS expression data (example: lines 147-158).

Any potential for sex differences in NO physiological and pathophysiological functions in podocytes? Include a clarification about the sex of cells/animals/humans when descripting data within the review. This is particularly relevant to established male-female differences in NO functions within the vasculature and renal tubule.

The role of NO in autoimmunity is highlighted within the abstract, but it is referred to very briefly within the review. Expand to highlight current knowledge on this critical area.

The methodology utilized for NO detection is referred to very briefly within figure 1 legend (line: 69). However, the detailed explanation of how the signal is detected is clarified later in lines 118-125. Reorganize to have the description of the detection method prior to the figure showing data.

It is not clear whether figure 1 is an original figure or adapted from a prior work

Figure 2: Consider differentiating lines/arrows that are established from hypothetical ones. This can be achieved by using solid vs. dotted lines/arrows.

Within figure 2: the authors use “iNOS” and “nNOS” designations for NOS isoforms, although the authors used the numerical designations earlier in the review article. For consistency, please use the most up to date nomenclature.

Minor comments:

Lines 36&37: “natriuresis” and “sodium excretion” are synonyms, remove one of the terms to avoid redundancy.

Line 209: change “nitric oxide” to “NO”

Reviewer 2 Report

In this review, the authors comprehensively describe the current knowledge on NO signalling in glomerular podocytes. The manuscript is well written and structured. Some aspects are illustrated with 2 figures. Therefore, only two minor suggestions should be addressed before this manuscript is suitable for publication.

  • Page 4, line 160: The authors state that “there are no data regarding the expression of specific NOS isoforms in parietal epithelial cells and mesangial cells”. I would not agree with this statement. The role of NOS2 as the main NO producing enzyme at least in mesangial cells has been extensively studied in rodent but also human mesangial cells. Please consider revising this phrase.
  • One or two more figures illustrating the signalling features of NO in podocytes would help the reader to understand this complex issue.
